# Introducing *MdTFL1* Promotes Heading Date and Produces Semi-Draft Phenotype in Rice

**DOI:** 10.3390/ijms241210365

**Published:** 2023-06-20

**Authors:** Van Giap Do, Youngsuk Lee, Seonae Kim, Sangjin Yang, Juhyeon Park, Gyungran Do

**Affiliations:** 1Apple Research Institute, National Institute of Horticultural and Herbal Science, Rural Development Administration, Gunwi 39000, Republic of Korea; seonaekim@korea.kr (S.K.); yangsangjin@korea.kr (S.Y.); wngus1113@korea.kr (J.P.); 2Planning and Coordination Division, National Institute of Horticultural and Herbal Science, Rural Development Administration, Wanju-gun 55365, Republic of Korea; microdo@korea.kr

**Keywords:** *TERMINAL FLOWER 1 (TFL1)*, early heading date, semi-draft, leaves angle

## Abstract

Flowering time (in rice, termed the heading date), plant height, and grain number are crucial agronomic traits for rice productivity. The heading date is controlled via environmental factors (day length and temperature) and genetic factors (floral genes). *TERMINAL FLOWER 1 (TFL1)* encodes a protein that controls meristem identity and participates in regulating flowering. In this study, a transgenic approach was used to promote the heading date in rice. We isolated and cloned apple *MdTFL1* for early flowering in rice. Transgenic rice plants with antisense *MdTFL1* showed an early heading date compared with wild-type plants. A gene expression analysis suggested that introducing *MdTFL1* upregulated multiple endogenous floral meristem identity genes, including the (early) heading date gene family *FLOWERING LOCUS T* and MADS-box transcription factors, thereby shortening vegetable development. Antisense *MdTFL1* also produced a wide range of phenotypic changes, including a change in overall plant organelles that affected an array of traits, especially grain productivity. The transgenic rice exhibited a semi-draft phenotype, increased leaf inclination angle, restricted flag leaf length, reduced spikelet fertility, and fewer grains per panicle. *MdTFL1* plays a central role in regulating flowering and in various physiological aspects. These findings emphasize the role of *TFL1* in regulating flowering in shortened breeding and expanding its function to produce plants with semi-draft phenotypes.

## 1. Introduction

Rice is an important crop worldwide. It is one of the most widely consumed staple foods. It is a primary source of nutrition for over half of the world’s population, especially in Asian countries such as China, India, and ASEAN nations. Rice plays a vital role in ensuring the food security of millions of people. Rice cultivation substantially affects the global economy. It is a major cash crop in many countries that supports the livelihoods of farmers, agricultural laborers, and numerous associated industries. Therefore, developing new rice varieties with elite properties, such as short-day (SD), high-yield, and high-quality (nutrients) rice, contributes to progressing toward achieving Goal 2 (zero hunger) of Sustainable Development.

The flowering of plants is primarily determined by endogenous genetic components and environmental cues. *TERMINAL FLOWER 1 (TFL1)* is well known and plays a key role in controlling flowering in plants. It belongs to a family of transcription factors (TFs), which are proteins that regulate gene expression by binding to specific DNA sequences. *TFL1* functions as a floral repressor that inhibits the transition from the vegetative phase to the reproductive phase in the shoot apical meristem (SAM) by suppressing the expression of genes involved in flower formation. *TFL1* maintains the indeterminacy of the meristem and prevents its conversion into a floral meristem [1,2]. The reduction or loss of function of *TFL1* promotes early flowering and has been aimedin many plants [3,4,5].

In plants, the gibberellin (GA) biosynthetic pathway comprises a complex network of enzymes encoded by multiple genes. These enzymes catalyze reactions that modify geranyl geranyl diphosphate (GGDP), resulting in the production of different GA compounds. GA20ox, a key enzyme in the biosynthesis of GAs, is involved in many catalytic reactions in the GA biosynthesis pathway [6]. GAs affect plant growth by influencing cell elongation and division, promoting the synthesis of cell-wall-degrading enzymes, and modulating gene expression. They play vital roles in stem elongation, leaf expansion, seed germination, and flowering time control [7,8,9,10,11]. GAs are also involved in fruit development and influence fruit size, seed development, and ripening [12,13,14]. Therefore, the loss of function or reduced activity of *GA20ox* genes restricts cell elongation, resulting in a semi-dwarf phenotype.

Plant height is one of the most important agronomic traits that directly affect rice grain yield and lodging tolerance. The Green Revolution occurred in the last century from the year 1960 to the 1970s, with the world wheat grain production increasing due to inheriting the advantages of producing new varieties of crops with potential agronomic traits such as shorter plant heights and increased grain yields. The semi-dwarf cultivar IR8 was developed in a high yield via the identified semi-dwarf (*sd-1*) gene in the Chinese variety Dee-geo-woo-gen (DGWG) at the International Rice Research Institute (IRRI) in 1967. A 383-base-pair deletion in the *GA20ox2* gene leads to a defective *gibberellin 20-oxidase (GA20)* gene that produces *sd-1* rice [6,15,16].

Manipulating the flowering time in crop plants such as rice is of great interest in agriculture. This allows for the development of varieties with optimized flowering characteristics, including early or late flowering, depending on specific environmental conditions or breeding objectives. For example, early flowering varieties can help extend the growing season in regions with short summers or enable synchronization with other crops in intercropping systems. Genetic engineering techniques such as antisense technology provide a means to target specific flowering time genes and modulate their expression, thereby altering the timing of flowering in plants [17,18,19,20]. By understanding the genetic basis of the regulation of flowering time and the underlying regulatory networks, researchers can develop strategies to manipulate this trait and breed crop varieties that exhibit the desired flowering characteristics.

The role of *TFL1* in regulating flowering has also been investigated. However, in addition to regulating flowering, its novel function in producing semi-dwarfism in rice using antisense expression of apple *MdTFL1* has not been well studied. We conducted this study to clarify the potential function of *TFL1* in triggering a broad range of phenotypic changes that affect crucial agronomic traits and yield productivity in rice. Herein, by utilizing antisense technology to specifically target *MdTFL1*, we not only provide a proof of concept for accelerating the flowering time in rice but also reveal that *MdTFL1* plays an essential role in various physiological aspects. We found that the ectopic expression of *MdTFL1* antisense in rice produces semi-dwarfs via restricted internode cell elongation related to GA synthesis. Moreover, we found that the antisense expression of *MdTFL1* led to the development of abnormal florets that reduced spikelet fertility in rice. The findings of this study have implications for crop improvement and plant development.

## 2. Results

### 2.1. Establishment of Expression Vector and Transgenic Plants

To investigate the role of *MdFTL1* in controlling flowering, we constructed a vector harboring antisense *MdTFL1* driven by the rice α-amylase 3D (RAmy3D) promoter. We obtained a transgenic plant harboring the plant expression vector 3D::MdTFL1 via in vitro tissue culture (Figure 1). The T-DNA region of this vector also included the *htpII* gene as a selection marker gene under the control of the 35S promoter (Figure 1B). The recombinant expression vector was then transformed into *Agrobacterium tumefaciens* LBA4404 for rice callus transformation. Following the embryonic rice callus transformation and in vitro tissue culture in a medium containing hygromycin B, putative transgenic rice seedlings were obtained (Figure 1A). The putative transgenic lines were screened at both stages (callus and seedling) during the establishment of transgenic plants via in vitro tissue culture. The calli and leaves of each candidate plant were sampled for a genomic DNA polymerase chain reaction (PCR). Successful insertion of the T-DNA region into the genome of the rice plants was evaluated based on the presence of the target gene (*MdTFL1*) and the selection marker hygromycin phosphotransferase (*htpII*). More than 16 rice callus cell lines were sampled for the genomic DNA PCR (Appendix A). After obtaining the putative transgenic seedlings generated from the transgenic calli, the leaves of these lines were confirmed again for absolute certainty of the establishment of the transgenic lines via plant genotyping (Figure 1C). The selected transgenic lines were transferred to soil and grown in a glasshouse for T0 seed harvesting (the seeds were collected in 2020).

### 2.2. Ectopic Expression of MdTFL1 Promotes Early Heading Date in Rice

The heading date was defined as the time from the sowing date to the emergence of the first panicle. The ectopic antisense expression of *MdTFL1* in the transgenic rice plants resulted in an early heading date compared with the heading date of the control plants (wild type (WT) and mock) (Figure 2). At the time of panicle initiation in the transgenic plants (3D:: MdTFL1), the control plants were still growing at the vegetative stage without any panicle formation (Figure 2A,B). The transgenic rice plants showed an average heading date of 59.7 ± 2.3 days after sowing (DAS). The average heading dates in the control plants were 80.6 ± 2.9 and 79.9 ± 3.7 DAS for the WT and mock plants, respectively (Figure 2C). There was no significant difference in the heading date between the WT and mock plants. However, the transgenic rice expressing *MdTFL1* promoted heading much earlier than the control plants by up to 20 days. Similar observations were made in the T0 and T1 plants (Appendix A). These results indicate that the expression of the *MdTFL1* gene was stable and the early heading phenotype was conserved in the transgenic rice plants.

### 2.3. Introducing of MdTFL1 Produced Semi-Dwarf Phenotype with Restrict Internode Elongation

Internode elongation was directly affected by the size of the cell in the elongation zone which, in turn, affected plant height. In this study, we found that the introduction of antisense *MdTFL1* into transgenic rice plants produced a semi-draft phenotype. The transgenic rice plants (3D:: MdTFL1) had much shorter overall heights than the WT plants (Figure 3A, Appendix A). The transgenic plant height was 34.6 ± 5.3 cm, and the WT plant height was 64.7 ± 4.4 cm. There was a significant difference in overall plant height for the same number of internodes. Therefore, the internode lengths of the transgenic plants were shorter than those of the WT plants, especially at the first (IN1) and second (IN2) internodes. The lengths of IN1 and IN2 were 17.9 ± 4.0 and 11.4 ± 2.1 mm in transgenic plants and 22.8 ± 4.5 mm and 15.8 ± 1.4 mm in WT plants, respectively. Furthermore, we measured the sizes of the cells in the elongation zone of the internode stem. The IN2 stems were examined at the cellular level under a microscope to uncover differences in internode length that led to differences in plant height between transgenic and WT plants. As shown in Figure 3B, there was a significant difference in cell size. The cell lengths of the transgenic plants were shorter, but the cell widths were greater compared with those of the WT. The cell length/width ratios were also significantly different between transgenic and WT plants, with a ratio of 3.3 in transgenic plants and a ratio of 4.5 in WT plants. With the same number of cell layers, the cells within the transgenic plants were thicker than those in the WT, leading to a thicker wall of the internode stem in the transgenic plants than in the WT plants. These observations were also made in the mock plants and were similar to the observations in the WT plants.

Furthermore, the expansion of cells in the elongation zone may be regulated by cell development (cell expansion and division) related to the genes involved in the GA synthesis pathway. We analyzed the expression of these genes, including the GA synthesis genes *OsCSP2* and *OsGA20ox3* and genes involved in cell expansion (*OsEXPA3* and *OsEX-PA4*). As shown in Figure 3C, the expression of these genes in transgenic plants was significantly lower than their expression in control plants; this difference occurred in parallel with a decline in cell expansion, leading to restricted internode elongation and the production of a semi-draft phenotype. We also evaluated the expression of other genes encoding for catalyze enzymes in the early GA biosynthesis pathway such as *OsCPS1* (catalyze GGDP to ent-CDP), *OsKO1*, and *OsKO2* (catalyze ent-Kaurene to GA–12); however, these genes were not significantly different in terms of gene expression when compared with the WT (Appendix A).

### 2.4. Modulated Inclination Angle of Leaves and Restricted Length of Flag Leaves

In addition to producing a semi-draft phenotype in transgenic plants, the introduction of *MdTFL1* produced plants with more inclined leaf angles, which affected the overall plant structure. Leaf angles were measured on the third and fourth leaves at 30 DAS. The leaf angles on the third and fourth leaves were 63.5° and 68.0° (transgenic rice plants) and 29.3° and 35.4° (WT rice plants), respectively (Figure 4A). Following the developmental stage, the leaf angles in the transgenic plants were continuously modulated by inclined leaves (45 and 90 DAS). As shown in the captured picture, the inclination angles of the leaves on the transgenic plants after 45 and 90 DAS were more than 90°. However, the inclination angles were significantly less in the WT plants (Figure 4A and Appendix A). Moreover, we observed differences in the lengths of the flag leaves. Compared with the WT, the transgenic rice plants produced flag leaves 10.3 cm shorter in length (Figure 4B). The lengths of the flag leaves in the WT and transgenic rice plants were 29.3 cm and 19 cm, respectively.

### 2.5. Reducing Spikelet Fertility and Affecting Grain Development Decreased the Number of Grains

The development of grain during the reproductive stage is crucial and directly affects rice yield. This process begins with panicle initiation and continues until grain filling in the milk stage. In this study, we observed that the ectopic expression of antisense *MdTFL1* in transgenic rice plants reduced spikelet fertility and affected grain development, resulting in fewer grains per panicle. As shown in Figure 5A, some spikelets (florets) in the panicles of the transgenic rice plants developed abnormally with opened lemmas, revealing internal reproductive organs such as anthers, stigmas, and ovaries. Abnormal spikelet development results in reduced spikelet fertility and failed grain and seed sets. In contrast, normal spikelet development is successful in seed settings. Moreover, the transgenic rice plants produced smaller panicles; thus, the panicles bore fewer grains than the WT plants (Figure 5B). The numbers of grains per panicle in the transgenic and WT plants were 63 ± 5.9 and 45 ± 3.2, respectively (Figure 5C). The upregulation of *OsGn1a* reduced the number of reproductive organs, resulting in decreased grain production (Figure 5D). In the comparison of grain size, we observed a significant difference in grain size between the transgenic and WT plants. The grain widths of the transgenic lines were thinner, but their lengths were greater than those of the WT. The ratio of grain length to width differed significantly between the transgenic and WT plants (ratios of 1.6 and 1.9, respectively). The grain weight of the WT lines was higher than that of the transgenic lines. The weights of 100 dehusked seeds from the WT and transgenic lines were 2.1 g and 1.4 g, respectively.

### 2.6. Expression Profile of Rice Endogenous Flowering-Related Genes

Introducing antisense *MdTFL1* promoted early heading by regulating the expression of flowering-related genes. A transcript expression analysis showed that endogenous floral regulator genes were upregulated in transgenic rice tissues. The expression of *MdTFL1* mRNA transcripts was evaluated using quantitative reverse transcription (qRT)–PCR and RT–PCR. The expression of *MdTFL1* mRNA transcripts in the transgenic lines varied. No *MdTFL1* mRNA was detected in the control lines (WT and mock) (Figure 6A). Moreover, a qRT–PCR analysis showed that in the transgenic lines, the introduced antisense *MdTFL1* upregulated the expression of endogenous floral regulator genes such as the heading date gene family *Heading date 1 (OsHd1), Early heading date 1 (OsEhd1), Heading date 3b (OsHd3b), Grain number, plant height, and heading date 7 (Ghd7),* and *RICE FLOWERING LOCUS T (RFT)* and MADS-box transcription factor (*OsMADS14, Os-MADS18,* and *OsMADS50*) (Figure 6B). We also evaluated the expression of these genes in callus cell lines. A similar phenomenon was observed, with the expression of most of these genes upregulated in the transgenic callus cell lines (Appendix A). However, there was no detectable expression of *OsEhd1* or *OsGhd7* in the callus cell lines. The expression of these two genes may be tissue- or developmental-stage-dependent.

## 3. Discussion

### 3.1. Introducing Antisense MdTFL1 Accelerates Early Heading in Transgenic Rice by Upregulating the Expression of Endogenous Flowering-Related Genes

In plants, flowering time is a critical developmental process that determines the transition from vegetative to reproductive growth. It is regulated by the complex interplay of genetic, environmental, and hormonal factors. The *FT* (*FLOWERING LOCUS T*)*/TFL* family of genes are floral regulators or flowering time genes that play important roles in regulating the transition from the juvenile phase to the reproductive phase and play opposite roles in this process. While *TFL* is a floral inhibitor that maintains the juvenile phase, *FT* is a floral inducer that plays a role in the transition from the vegetative to the production phase. Previous studies have presented compelling evidence that *MdTFL1* downregulation using antisense technology leads to accelerated flowering in transgenic apples and tobacco [3,21]. In contrast, *MdTFL1* overexpression reduced the transition from the vegetative to the reproductive phase in transgenic Arabidopsis [22]. In this study, we observed that the introduction of antisense *MdTFL1* accelerated the early heading date in transgenic rice.

In many plant species, including rice, flowering time is influenced by various genes involved in the flowering time control pathways [23]. These genes interact in complex regulatory networks to integrate internal and external cues, such as the photoperiod (day length), temperature, hormonal signals, and developmental factors, to initiate flowering at appropriate times. Altering the expression of one of them may disturb the expression of the others. In this study, we used antisense technology to suppress *MdTFL1* expression in transgenic rice. The integration of antisense *MdTFL1* into the rice genome has been shown to regulate the expression of floral integrator genes which ultimately initiate the transition from vegetative growth to flowering. We evaluated the expression of endogenous flowering-related genes to investigate the role of MdTFL1 in the regulation of rice heading. The expression of the endogenous floral genes *OsHd1*, *OsEhd1*, *OsHd3b*, *RFT*, *OsMADS14*, *OsMADS18*, and *OsMADS50* was upregulated and *OsGhd7* was downregulated in the transgenic rice plants (Figure 6B). Additionally, the upregulated expression of most of these genes was observed in transgenic rice calli (Appendix A).

*Ehd1* promotes the expression of downstream florigen genes, such as *Hd3a* and *RICE FLOWERING LOCUS T1 (RFT1*), under both long-day (LD) and SD conditions by integrating them into different upstream florigen genes [24,25]. Their mobile flowering signals move from the leaves to the shoot apical meristem and are essential for rice flowering [26]. The upregulation of *Ghd7* not only delays heading but also increases plant height and panicle size under LD conditions [27]. *Hd3* (*Hd3a* and *Hd3b*), homologs of *FLOWERING LOCUS T (FT)* in Arabidopsis, are quantitative trait loci (QTL) localized on chromosome 6 and are photoperiod-sensitive heading date genes. An analysis of expression in response to photoperiod revealed that *Hd3a* promotes heading under SD conditions, and *Hd3b* causes late heading under LD conditions [28,29]. However, in this study, we observed that *Hd3a* was not detected (Appendix A), whereas *Hd3b* was upregulated in 3D::MdTFL1 transgenic rice under LD conditions (Figure 6B).

The MADS-box family encodes conserved transcription factors that control the transition from vegetative to reproductive growth. the overexpression of *OsMADS15*, an ortholog *APETALA1 (AP1)* gene in Arabidopsis, results in early flowering and reduced plant height in transgenic rice [30]. In another study, the overexpression of *RFT1* induced extremely early flowering, bypassing normal vegetative development to directly form spikelets from rice calli. MADS-box genes (*OsMADS14, OsMADS15,* and *OsMADS18*) were upregulated in the shoot apical meristem (SAM) of the transgenic lines [31]. In our study, we found that the MADS-box genes *OsMADS14, OsMADS18*, and *OsMADS50* were highly induced in both the seedling leaves and calli of the transgenic rice (Figure 6B and Appendix A). Taken together, the expression of *MdTFL1* antisense promoted an early heading date in transgenic rice by stimulating the endogenous regulatory genes associated with reproductive development (rice florigen genes), thereby regulating their expression.

### 3.2. MdTFL1 Antisense Produces Semi-Dwarf Phenotype by Negatively Regulating GA Biosynthesis Genes

As expected, introducing *MdTFL1* antisense resulted in shortened vegetative phase via the acceleration of the heading date. Surprisingly, 3D::MdTFL1 transgenic rice plants exhibiting distinctive phenotypes were also observed in this study. The most distinctive phenotype exhibited was semi-dwarfism in the 3D:: MdTFL1 transgenic rice plants (Figure 2A and Figure 3A). Its connection to the GA biosynthesis pathway suggests that the downregulation of *MdTFL1* via antisense expression may affect the synthesis or signaling of GAs. In this study, we found that GA-biosynthesis-related genes, *OsCPS2* and *OsGA20ox3*, were reduced in the transgenic rice, leading to the reduced synthesis of GAs. Moreover, we found that the expression of the cell-elongation-related genes *OsEXPA3* and *OsEXPA4* was downregulated in transgenic rice (Figure 3C). Reducing GA-biosynthesis-related genes (*OsCPS2* and *OsGA20ox3*) might cause the downregulation of cell-elongation-related genes, effectively shortening cell lengths and inevitably reducing plant heights (Figure 3A,B). Although the cell lengths of the transgenic plants were shorter than those of the WT plants, the cell widths were greater. This indicates that the cell walls of the transgenic lines became thicker. These characteristics, together with semi-dwarfism supporting lodging tolerance, contribute to reducing the damage caused by wind and rain. Lodging tolerance is a critical agronomic trait. It is a helpful characteristic that can reduce the lost rice yield, especially in countries with wet rice agriculture, such as the Philippines, Indonesia, and Vietnam, which often suffer from many storms during the rice-growing season.

The ectopic expression of *OsEATB* (a rice ethylene-response AP2/ERF factor) restricts internode elongation by reducing the expression of the GA biosynthetic genes *OsCPS2* and *GA20ox* but not *OsCPS1* and *OsKO2* [32]. These observations were also made in our study, as described in Section 2.3. The loss of function of *FLOWERING LOCUS T2* (*FT2)* induces internode elongation by upregulating the expression of *GA3ox2* via the 13-hydroxylation gibberellin biosynthesis pathway in polar plants under LD conditions [33]. This suggests that the introduction of floral genes into plant cells may alter the expression of other genes in the GA biosynthesis pathway, thereby affecting internode elongation and plant height.

Grain yield is one of the most important agronomic traits directly determining rice production. Grain yield is controlled by several genes known as QTLs. Crop breeders have been working to improve the grain yield using genetic approaches. Grain yield can be affected by multiple factors such as grain size, grain weight, and number of grains per panicle. Grain size is regulated by the QTL *GRAIN SIZE3 (GS3)* [27,34]. An upregulation of *Ghd7* has been reported that functions to increase the number of grains per panicle, resulting in a delayed heading date and increased rice height [27]. The grain number is a QLT controlled by *Grain number 1a (Gn1a)* [35]. The downregulation or loss of function of *Gn1a* increases the grain number, thereby increasing grain yield [35,36]. In our study, the transgenic rice plants produced small panicles and long grains with lighter weights than the control plants (Figure 5B,C,E,F). These throughputs may be due to abnormalities in the development of spikelets with opened lemmas that reduce spikelet fertility, thus decreasing the productivity yield potential (Figure 5A). Similar observations of abnormal spikelet development have been made in transgenic rice and wheat [37,38,39,40,41].

In summary, the introduction of antisense *MdTFL1* upregulates multiple endogenous floral meristem identity genes, resulting in an early heading date. Moreover, the antisense expression of *MdTFL1* produced distinctive phenotypic changes that exhibited a semi-draft with inclined leaf angles, shorter flag leaves, abnormal spikelets, and fewer grains per panicle. These distinctive changes in phenotype were a result of the interaction of many genes involved in floral gene regulation, GA-biosynthesis-related genes, and the crosstalk between these pathways (Figure 7). This indicates that the *MdTFL1* gene not only plays a central role in regulating flowering but also plays an important role in various physiological processes.

## 4. Materials and Methods

### 4.1. Construction of Expression Vector and Agrobacterium Transformation

For vector construction, *MdTFL1* was isolated and amplified from the apple mRNA using a specific primer set (MdTFL1-F: GGGGTACCATGAAAAGAGCCTCGGAG and MdTFL1-R: CGGGATCCCTAGCGTCTTCTAGCTG, where underlined sequences are KpnI and BamHI enzyme sites, respectively). *MdTFL1* was cloned into the plant expression vector pMYD319 (pCAMBIA1300-based backbone) under the control of the rice α-amylase 3D (RAmy3D) promoter [42] using restriction enzyme sites (BamHI–KpnI). The recombination vector was transformed into *Escherichia coli* DH5α competent cells and the plasmid DNA of the recombination vector was confirmed via BamHI–KpnI enzyme digestion. The resulting construct, 3D::MdTFL1, was introduced into *Agrobacterium tumefaciens* LBA4404 using the triparental mating method [43].

### 4.2. Plant Materials and Growth Conditions

Mature seeds of rice (*Oryza sativa* L. Samkwang) were dehusked, surface-sterilized, and then cultured as previously described [44] with slight modifications. Briefly, the seeds were surface-sterilized with 70% ethanol for 5 min, treated with a solution of 2.5% NaOCl and 0.01% Tween20 for 15 min, and finally washed with sterile water several times. The sterilized seeds were germinated on the embryonic calli induction medium N6CI (4 g/L of N6 medium salts (Duchefa, Haarlem, The Netherlands), 30 g/L of sucrose, and 2 mg/L of 2,4-dichlorophenoxyacetic acid (2,4-D), 0.02 mg/L of kinetin, and 2.3 g/L of phytagel, pH 5.7). After two weeks, the embryonic calli were detached from the seedlings and cultured for 10 days on the N6CI medium before the transformation.

### 4.3. Plant Transformation and the Establishment of Transgenic Plants

Overnight cultures of *A. tumefaciens* LBA4404 cells were collected and resuspended in liquid N6CI (without phytagel) for rice callus transformation via the *Agrobacterium*-mediated transformation, as previously described [42]. Transgenic rice calli were obtained in a selection medium containing 50 mg/L of hygromycin B (N6SE = N6CI + 50 mg/L of hygromycin B + 250 mg/L of carbenicillin). Transgenic rice calli were propagated and subcultured every 3–4 weeks on N6SE.

To generate rice seedlings, the transgenic rice calli were then transferred and cultured on shoot regeneration medium MSSR (4.3 g/L of MS vitamins, 30 g/L of sucrose, 30 g/L of sorbitol, 5 mg/L of kinetin, 1 mg/L of NAA, 4 g/L of gelrite and supplemented with 50 mg/L of hygromycin B). After 4–6 weeks of culturing on MSSR, the generated shoots were cultured on rooting medium MSR (4.3 g/L of MS vitamins, 30 g/L of sucrose, 0.5 mg/L of IBA, 3 g/L of gelrite and supplemented with 50 mg/L of hygromycin B) for the establishment of the whole transgenic seedling, including the root, before sowing into the soil.

The in vitro culture took place in a culture room with controlled conditions: a 16 h/8 h (light/dark) photoperiod with a light intensity of 100 μM/m^2^ s^1^ and a temperature maintained at 25 °C. The rice plants in the soil were grown in a glasshouse under LD conditions, maintaining a temperature of 32/25 °C (day/night) during the warm periods of the year (from April to September).

### 4.4. Plant Genotyping

The calli and leaves of the putative transgenic rice were sampled for double confirmation of the selected transgenic rice using genomic DNA PCR with specific primer sets (Appendix A). The correct transgenic calli/plant were confirmed using two specific primer sets for the amplification of the target gene *MdTFL1* and the selection gene *hptII*. Genomic DNA was extracted using a DNeasy^®^ Plant Mini Kit (Cat. #69204; Qiagen, Hilden, Germany). PCR was performed using the Maxime™ PCR PreMix (i-StarTaq) (Cat. #25167; Intron, Seongnam, Korea), and the products were visualized using the Image Lab^TM^ program (Bio-Rad Laboratories, Inc., Hercules, CA, USA)

### 4.5. RNA Isolation, cDNA Synthesis, and Gene Expression Analysis

For gene expression analysis, the calli and leaves of the transgenic rice plants were sampled for RNA isolation. Total RNA isolation and cDNA synthesis were performed as described previously [21]. Transcript expression was quantified using RT–PCR and RT–qPCR. The RT–PCR was carried out in 20 μL of reaction volume using a Maxime™ PCR PreMix with the following run protocol: initial denaturation at 95 °C for 5 min, followed by 30 repeated cycles of denaturation/annealing/extension (95 °C/58 °C/72 °C for 15 s of each step) and a final extension at 72 °C for 5 min. The PCR products were separated on a 0.8% agarose gel via electrophoresis, and the bands were visualized using the Image Lab^TM^ program (Bio-Rad Laboratories, Inc.). The qRT–PCR was performed on a LightCycler 480 II Real-Time PCR System (Roche Diagnostics, Mannheim, Germany) using a LightCycler 480 SYBR Green I Master Mix (Roche). The qRT–PCR was carried out in 20 μL of reaction volume with 50 μg of diluted cDNA and specific primer sets (Appendix A). Transcript expression was calculated via normalization to the reference gene *OsUbi1*.

### 4.6. Plant Phenotyping

The heading date was scored when the first panicle emerged and was expressed as days after sowing (DAS). The plant height and internode length were measured during the ripening phase (mature stage). The plant height was measured from the base of the soil to the panicles. The internodes were numbered from 1 to 5 (from the tip to the root). The leaf angle was measured at the vegetative phase for the third and fourth leaves using a protractor and expressed in degrees (°). The flag leaf length was measured after heading. Grain developmental morphology was observed at the milk stage, and grain productivity (number of grains per panicle, weight, and size) was evaluated at the mature stage after harvesting. Plant phenotyping was collected from the recorded data of planted rice plants over 3 years (from 2020 (T0) to 2022 (T2)).

### 4.7. Light Microscopy

For the stem elongation measurements, the second internode stems in the vegetative phase (before the panicle initiation stage) were sampled and performed as previously reported [21]. Cell images were obtained using a light microscope (Axioskop 2; Carl Zeiss, Oberkochen, Germany). The cell size was analyzed using Olympus cellSens Standard software (Olympus, Shinjuku, Tokyo, Japan).

### 4.8. Statistical Analysis

Tukey’s test was used to analyze significant differences between multiple groups, and a one-way ANOVA was used to analyze significant differences between pairwise groups. The results are expressed as means ± standard deviations from three biological replicates, with *p* < 0.05 indicating significant differences.

## 5. Conclusions

In this study, we successfully established transgenic plants using plant transformation and in vitro tissue culture techniques. Through our investigation into the identification of genes involved in controlling flowering and their effects on agronomic traits, we found that the overexpression of apple *MdTFL1* resulted in an early heading date via the disruption of the expression of endogenous floral regulator genes. Specifically, we observed the upregulation of the heading date gene family (*OsHd1*, *OsEHD1*, and *OsHd3B*), *FLOWERING LOCUS T* (*RFT*), and MADS-box transcription factors (*OsMADS14, OsMADS18,* and *OsMADS50*). Furthermore, the introduction of antisense *MdTFL1* significantly affected various phenotypic traits in transgenic rice. The transgenic plants exhibited a semi-dwarf phenotype with a reduced plant height, altered leaf angle, and a decreased number of grains per panicle. Based on our findings, we concluded that floral genes can be used to expedite the flowering process. Moreover, the transgenic approach employed in this study showed its potential as a valuable strategy for shortening breeding programs. These results provide important insights into the manipulation of flowering control genes and their application in crop improvement, thereby contributing to the advancement of agricultural practices. Although the transgenic rice demonstrated advantages such as early heading (short-day cultivation) and semi-dwarfism (beneficial in lodging tolerance), the transgenic line exhibited a wide range of phenotypic changes, especially the development of abnormal florets that reduced the grain yield potential in the rice. Therefore, it is imperative to consider that in addition to attaining the desired traits, unforeseen traits may arise.

## Figures and Tables

**Figure 1 ijms-24-10365-f001:**
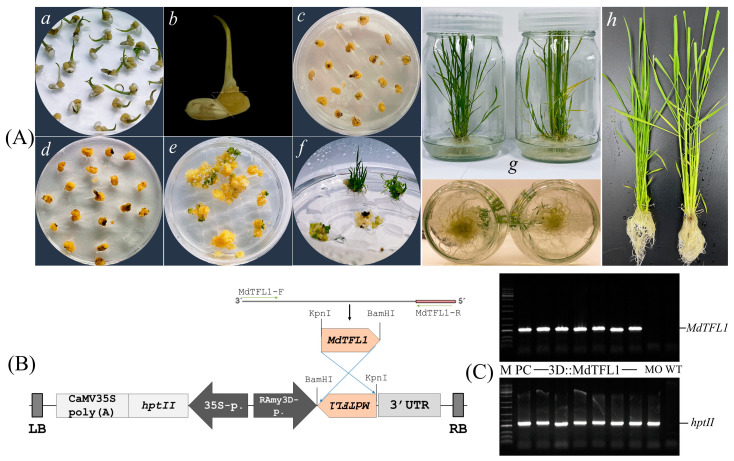
The stepwise establishment of transgenic rice. (**A**) The transgenic rice seedlings were established using Agrobacterium-mediated transformation via in vitro tissue culture. (**a**) Callus induction of dehusked seeds on callus induction medium N6CI; (**b**) embryonic calli were isolated for rice callus transformation via Agrobacterium-mediated transformation; (**c**) co-culture of Agrobacterium and rice embryonic calli; (**d**–**e**) selection of putative transformed calli on selection medium N6SE; (**f**) growth of putatively transgenic lines on shooting medium MSSR; (**g**,**h**) growth of transgenic seedlings on rooting medium MSR. (**B**) Schematic representation of the T-DNA region harboring the target gene *MdTFL1* and selection marker gen *hptII* under control of the rice α-amylase 3D (RAmy3D) promoter and the cauliflower mosaic virus 35S (CaMV35S) promoter, respectively. (**C**) Genomic DNA PCR of MdTFL1 and hptII in rice leaf tissues. Lane M, 1 Kb plus DNA ladder marker. Lane PC, plasmid DNA of 3D::MdTFL1 as a positive control; lanes 3D::MdTFL1, independent 3D::MdTFL1 transgenic callus lines; lane MO, genomic DNA of transgenic rice callus transformed with vector pMYD319 used as a mock; lane WT, genomic DNA of wild-type (non-transgenic callus) used as a negative control.

**Figure 2 ijms-24-10365-f002:**
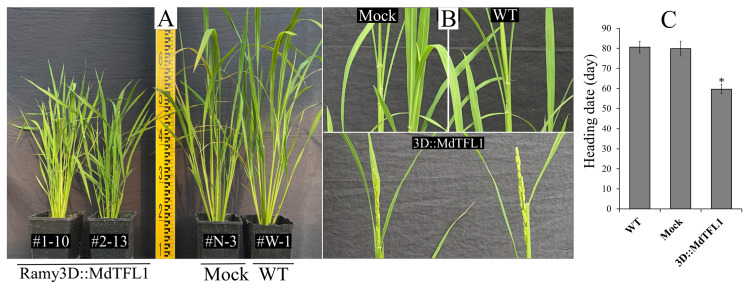
Comparison of heading dates between the 3D::MdTFL1 and control (mock and WT) rice plants. (**A**,**B**) 3D::MdTFL1 transgenic rice lines showed early heading at 59.7 DAS, while the control (mock and WT) lines remained in the vegetative stage. (**C**) Evaluation of heading dates of the transgenic lines and control lines. Data were evaluated from T2 plants (cultivated in the year 2022) germinated from seeds of T1 plants (cultivated in the year 2021) and grown in a glasshouse under LD conditions. Data are presented as means ± SDs (n ≥ 20). The asterisks indicate significant differences compared to the control (*p* < 0.05).

**Figure 3 ijms-24-10365-f003:**
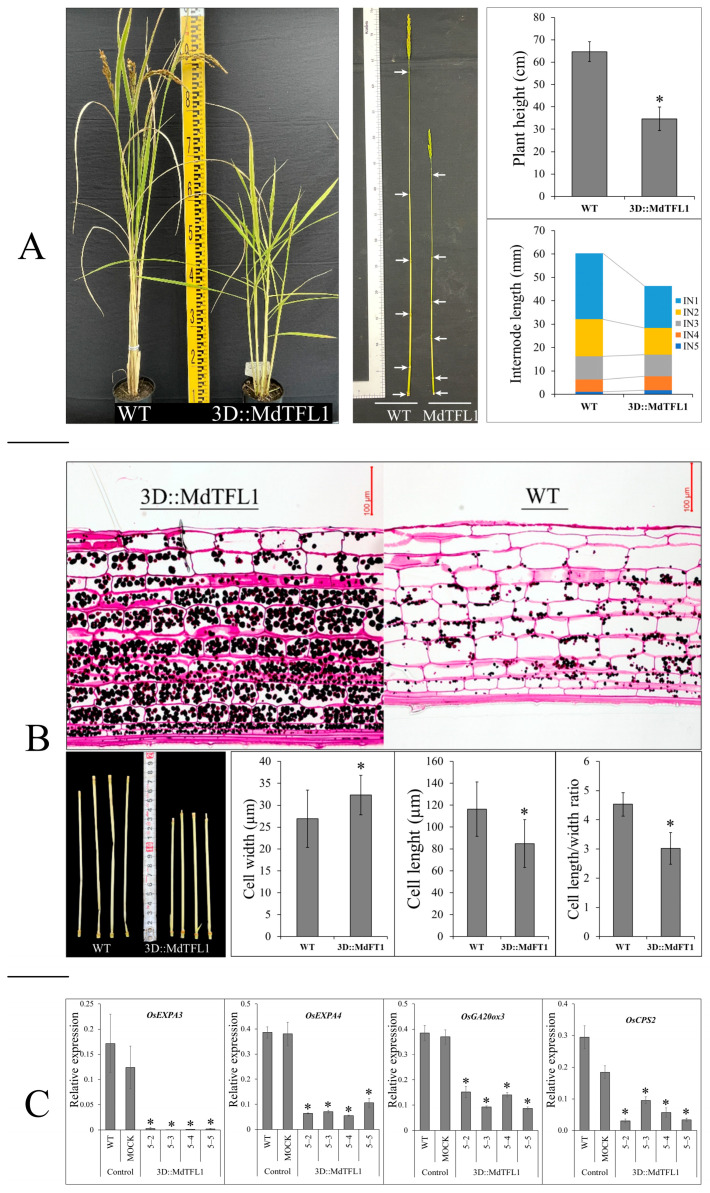
Ectopic expression of *MdTFL1* antisense produced a semi-dwarf phenotype. (**A**) Comparisons of plant height between the 3D::MdTFL1 and control (mock and WT) rice plants with differences in internode length. The 3D::MdTFL1 transgenic rice plants developed with shorter total heights and internode lengths. (**B**) Cell size in the elongation zone (IN2) under microscopy. (**C**) Expression of genes involved in GA synthesis genes (*OsCSP2* and *OsGA20ox3*) that affect cell expansion (*OsEXPA3* and *OsEX-PA4*) restricted internode elongation that led to differences in plant height. The asterisks indicate significant differences compared to the control (*p* < 0.05).

**Figure 4 ijms-24-10365-f004:**
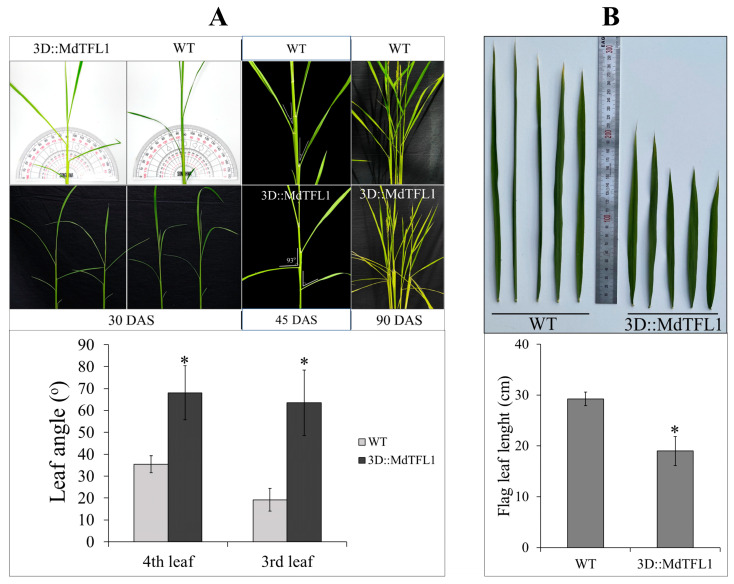
Effect of transgenic 3D::MdTFL1 on leaf angles and flag leaf length. Ectopic expression of *MdTFL1* antisense modulated inclination leaf angle (**A**) and restricted flag leaf length (**B**). Data are presented as means ± SDs (n ≥ 20) and were evaluated from recorded data from over the last 3 years of T0 to T2 plants. The asterisks indicate significant differences compared to the control (WT and mock) (*p* < 0.05).

**Figure 5 ijms-24-10365-f005:**
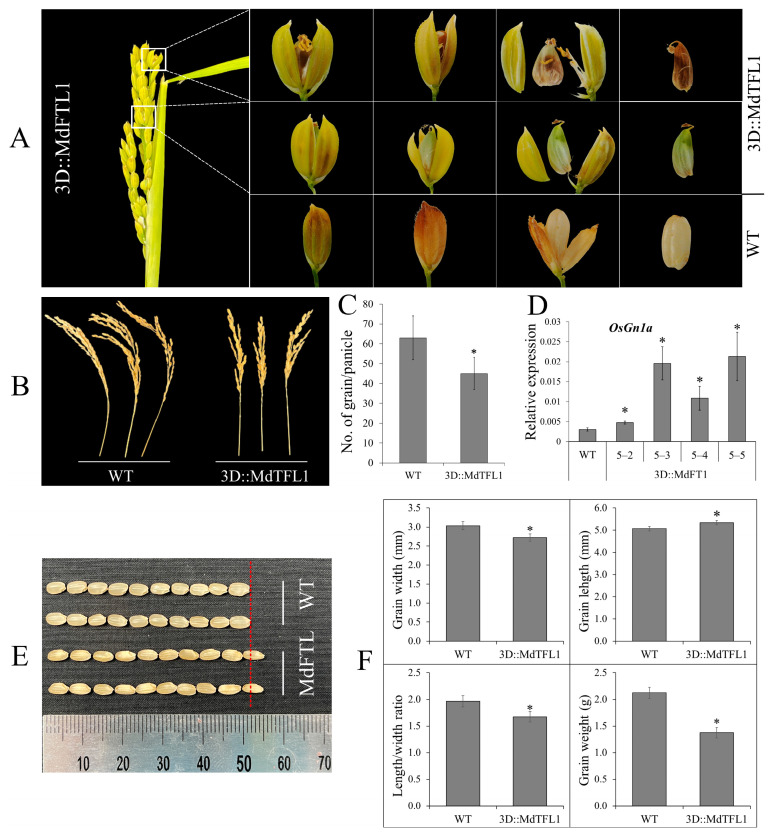
Effect of antisense expression of *MdTFL1* on spikelet/grain development and yield potential. (**A**) Abnormal development of grain that reduced spikelet fertility caused failed setting grain (brown grain) or less grain filling (lighter seed weight). (**B**) The panicles of WT and transgenic lines at the mature stage of seed production. (**C**) Number of grains per panicle for WT and transgenic lines. (**D**) Expression of *OsGn1a* gene defined grain number. (**E**,**F**) Grain size and weight. Values are means ± SDs (n ≥ 20). The asterisks indicate significant differences compared to the WT (*p* < 0.05).

**Figure 6 ijms-24-10365-f006:**
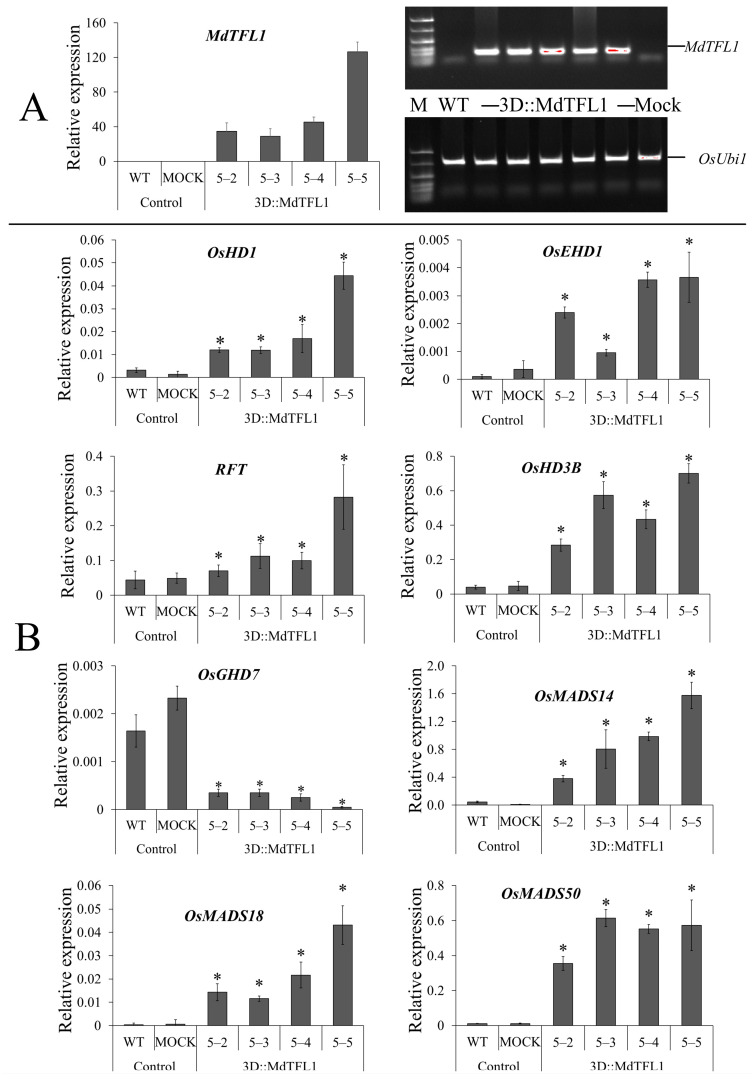
Expression profile of endogenous flowering-related genes in rice plants. The expression levels of *MdTFL1* (**A**) and rice flowering-related genes (**B**) were normalized to *OsUbi1*. Data represent the means ± SDs from three biological replicates. The asterisks indicate significant differences compared to the control (WT and mock) (*p* < 0.05).

**Figure 7 ijms-24-10365-f007:**
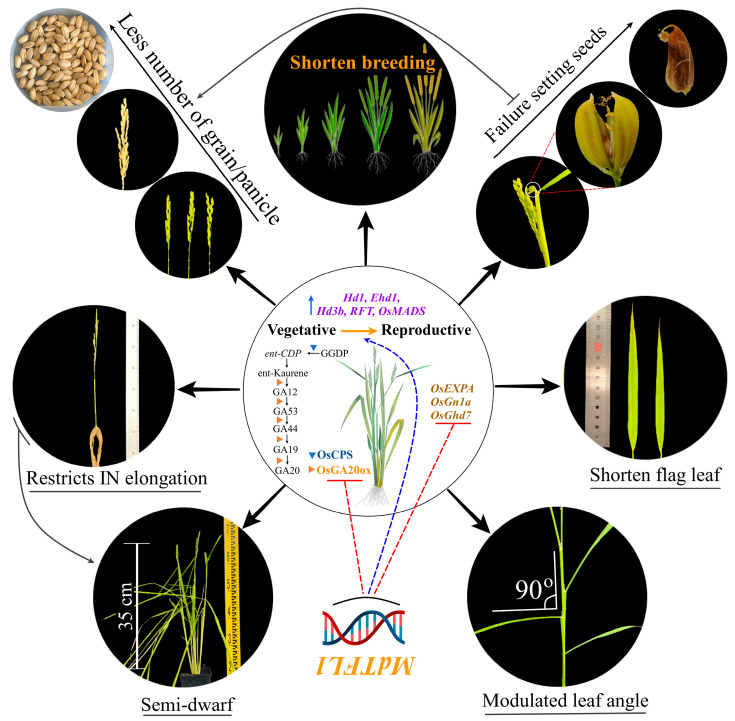
A model scheme showing that introducing antisense *MdTFL1* accelerates early heading in transgenic rice by upregulating the expression of endogenous flowering-related genes and producing a semi-dwarf phenotype by negatively regulating the GA biosynthesis pathway. The expression of *MdTFL1* antisense triggered the expression of multiple endogenous genes in rice, resulting in a wide range of phenotypic changes. The upside-down representation of *MdTF1* indicates that the reverse direction (antisense) of *MdTF1* was introduced into the rice plant.

## Data Availability

Data are contained within the article or Supplementary Material.

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
