# Peer review of "Introducing MdTFL1 Promotes Heading Date and Produces Semi-Draft Phenotype in Rice"

_ijms, 2023, doi:10.3390/ijms241210365_

Round 1
Reviewer 1 Report
1. Manuscript title: I don't understand the logic of the title and I think it needs to be rewritten. The authors need to understand that "phenotype" and "phenotyping" is different, also, "high-throughput" is misused.
2. L11: The authors need to define more accurately "heading date", and need to understand that it is somehow different from "flowering time". It's very critical to further molecular and genetic investigations in a precise way.
3. L14: Avoid using first-person writing throughout the manuscript.
4. L20: Don't understand what you mean by"high-throughput phenotyping".
5. L25: Don't understand what you mean by "shortened breeding".
6. Keywords: "gene expression" is too normal
7. Figure 7: "MdTFL1" in the middle is upside down.
8. L450: How many replicates were conducted in each treatment?
9. L466-471: MdTFL1 can reduce flowering time but it induced various abnormal phenotypes, so I think the conclusion should be more conservative.
Reviewer 2 Report
TFL1 is well known and plays a key role in controlling flowering in plants. TFL1 functions as a floral repressor that inhibits the transition from the vegetative to reproductive phase in the shoot apical meristem by suppressing the expression of genes involved in flower formation, including GA synthesis genes and genes involved in cell expansion. The reduction or loss-of-function of TFL1 promotes early flowering. Authors used antisense technology to suppress of apple MdTFL1 expression for early flowering in rice. Transgenic rice plants with antisense MdTFL1 showed an early heading date compared with wild-type plants. The authors showed that MdTFL1 plays a central role in regulating flowering and in various physiological aspects. The authors showed that the overexpression of apple MdTFL1 resulted in an early heading date by disrupting the expression of endogenous floral regulator genes. Furthermore, the introduction of antisense MdTFL1 significantly affected various phenotypic traits in transgenic rice, resulting in fewer grains per panicle. Based on our findings, the authors concluded that floral genes can be used to expedite the flowering process.
Research methods are clearly stated, they can be reproduced. The correct logic of setting up experiments is traced.
The manuscript is well illustrated, there are many photographs that help to quickly understand the work, and are also an adornment of the study.
L. 51. GGDP - must be deciphered at the first mention.
L. 79-89. The introduction should end with the purpose of the study, this is not included here.
L. 81. It must be indicated that the MdTFL1 gene was taken from an apple tree.
L. 186 and L. 296. The authors write "results not shown", although these results could have been placed in Supplementary Materials.
L. 326-327 The authors write that "These characteristics, together with semi-dwarfism supporting lodging tolerance, contribute to reducing the damage caused by wind and rain."
However, it is known that “The survival and emergence of rice under flood stress in the living environment can be ensured by cultivating flood-resistant varieties that exhibit vigorous shoot growth. Due to the fast mobilization of nutrients, the shoots spread quickly and break free from the water’s surface in need of oxygen.”
The conclusion suggests itself that the transgenic rice lines obtained by the authors cannot be used in agriculture. What is the opinion of the authors?
Round 2
Reviewer 1 Report
I don't have further questions.